# BBX24 Increases Saline and Osmotic Tolerance through ABA Signaling in Arabidopsis Seeds

**DOI:** 10.3390/plants12132392

**Published:** 2023-06-21

**Authors:** Tai S. Chiriotto, Maite Saura-Sánchez, Carla Barraza, Javier F. Botto

**Affiliations:** Instituto de Investigaciones Fisiológicas y Ecológicas Vinculadas a la Agricultura (IFEVA), Consejo Nacional de Investigaciones Científicas y Técnicas (CONICET), Facultad de Agronomía, Universidad de Buenos Aires (UBA), Ciudad Autónoma de Buenos Aires C1417DSE, Argentinasaurasanchez@agro.uba.ar (M.S.-S.);

**Keywords:** germination, drought, saline stress, B-box transcription factor, BBX24

## Abstract

Seed germination is a critical stage for survival during the life cycle of an individual plant. Genetic and environmental cues are integrated by individual seeds to determine germination, mainly achieved through regulation of the metabolism and signaling of gibberellins (GA) and abscisic acid (ABA), two phytohormones with antagonistic roles. Saline and drought conditions can arrest the germination of seeds and limit the seedling emergence and homogeneity of crops. This work aimed to study the function of BBX24, a B-Box transcription factor, in the control of germination of *Arabidopsis thaliana* seeds imbibed in saline and osmotic conditions. Seeds of mutant and reporter GUS lines of BBX24 were incubated at different doses of NaCl and polyethylene-glycol (PEG) solutions and with ABA, GA and their inhibitors to evaluate the rate of germination. We found that BBX24 promotes seed germination under moderated stresses. The expression of *BBX24* is inhibited by NaCl and PEG. In addition, ABA suppresses BBX24-induced seed germination. Additional experiments suggest that BBX24 reduces ABA sensitivity, improving NaCl tolerance, and increases GA sensitivity in seeds imbibed in ABA. In addition, BBX24 inhibits the expression of *ABI3* and *ABI5* and genetically interacts upstream of HY5 and ABI5. This study demonstrates the relevance of BBX24 to induce drought and salinity tolerance in seed germination to ensure seedling emergence in sub-optimal environments.

## 1. Introduction

Drought and saline stresses are a global change phenomenon that seriously limits crop production and distribution. These stresses are gradually becoming more severe in many places, mainly in arid or semi-arid areas, due to climate change [1] Given that plants are mostly sessile and thus unable to relocate, they have evolved complex morphological, physiological, cellular and molecular adaptations to cope with those stresses [2,3].

Germination is defined as the developmental process that begins with hydration of the seed tissues and culminates when the radicle emerges [4,5,6]. Germination is determined by individual seeds integrating genetic and environmental cues. This is mainly achieved through regulation of the metabolism and signaling of gibberellins (GA) and abscisic acid (ABA), two phytohormones with antagonistic roles. Germination is described via three phases of water uptake. Initially, the seed imbibes and reinitiates metabolic processes in phase I; subsequently, it enters a lag phase of water uptake during phase II, which is followed by further water uptake, resulting in the emergence of the radicle in phase III [7]. During imbibition, ionic and osmotic solutions can reduce germination either by limiting water absorption by the seeds, by affecting the mobilization of stored reserves, or by directly affecting the structural organization or synthesis of proteins in germinating embryos [8].

Under osmotic stress conditions such as drought and high salinity, ABA is accumulated and numerous genes functioning in stress response and tolerance are induced. Osmotic stress-responsive genes are regulated by ABA-dependent and ABA-independent pathways [2,3]. An environment that is unfavorable for seed germination leads to the activation of the ABA biosynthesis genes, which results in a higher content of the active ABA pool and the upregulation of ABA-responsive genes [9]. ABA INSENTIVE 5 (ABI5) transcription factor is a central and master node inducing ABA response in seeds [10,11]. *ABI5* expression is strongly induced by ABA, water stress and high salinity. Some transcription factors, such as ELONGATED HYPOCOTYL 5 (HY5), can bind to and activate ABI5 promoter. HY5 loss-of-function mutant seeds showed a lower sensitivity to ABA, salinity and water stress in germination [12].

B-box-containing zinc finger transcription factors (BBX) mediate transcriptional regulation and protein–protein interactions in plant growth and development, often integrating environmental information with the hormone signaling network to cope with stress conditions [13,14,15]. Several BBX proteins have been documented to be involved in seedling de-etiolation, synthesis of anthocyanins, shade avoidance, flowering and responses to abiotic and biotic stresses [13,16]. BBX24 has been documented to inhibit seedling de-etiolation [17,18,19,20], repress UVB signaling [21,22] and promote shade avoidance [17,23,24] and flowering [25]. BBX24 was originally isolated as a SALT-TOLERANT (STO) protein in an Arabidopsis cDNA clone screen that conferred increased salt-tolerance in *Saccharomyces cerevisiae* [26,27]. BBX24 overexpression increases salt tolerance in Arabidopsis [27]. It has been shown that BBX19 and BBX21 regulate the germination of Arabidopsis seeds [28,29]. BBX19 suppresses germination by enhancing ABA hypersensitivity through a *cis*-regulation on the ABI5 promoter, which induces its expression [28]. On the other hand, *bbx21* mutant seeds are hypersensitive to ABA, suggesting that BBX21 induces germination through ABA signaling [29]. BBX21 can regulate HY5 at the post-transcriptional level by interfering with HY5 binding on the *ABI5* promoter and can directly interact with ABI5 to inhibit its expression during seed germination in ABA [29].

However, the function of BBX24 in seed germination is completely unknown. Regarding the relevance of BBX24 increasing salt tolerance [26,27], this work aimed to study the function of BBX24 in the control of germination during the imbibition of Arabidopsis seeds in saline and osmotic stress. We found that BBX24 promotes seed germination under moderated stresses. The expression of *BBX24* is inhibited by NaCl and polyethylene-glycol (PEG) imbibition solutions. In addition, ABA suppresses BBX24-induced seed germination. Our experiments suggest that BBX24 can reduce ABA sensitivity, increasing the tolerance to NaCl. BBX24 also increases GA sensitivity when seeds are imbibed in ABA solution. At the molecular level, BBX24 inhibits the expression of *ABI3* and *ABI5* in ABA, and BBX24 genetically interacts upstream of HY5 and ABI5. Altogether, this study investigates the relevance of BBX24 in the regulation of seed germination in sub-optimal environments.

## 2. Results

### 2.1. BBX24 Promotes Seed Germination in Drought and NaCl

We designed an experiment to evaluate seed germination in water, drought and saline conditions. Firstly, we established the saline and drought conditions simulated by the addition of NaCl and PEG in the imbibition medium. We selected 150 µmol NaCl and 0.4 MPa to obtain ~50% inhibition of Col-0 seeds (Appendix A. Then, we sowed Col-0, *bbx24-1* and *BBX24ox* seeds in water, 150 mM NaCl and −0.4 MPa and counted germination every day until the end of the experiment at 7 days. In the water condition, we did not find significant differences in germination between Col-0, *bbx24-1* and *BBX24ox* lines at the end of the experiment. However, the rate of germination was different between genotypes: *bbx24-1* seeds delayed germination and *BBX24ox* accelerated germination compared with Col-0 at 48 h after exposition to light (Figure 1). Col-0 and *BBX24ox* seeds germinated at ~40% in NaCl and PEG solutions at the end of the experiment, while *bbx24-1* seeds germinated at ~10% in both stress conditions. Significant differences in germination between Col-0 and *bbx24-1* seeds were observed after 48 h or 72 h of light exposure in PEG or NaCl, respectively.

### 2.2. PEG and NaCl Inhibit the Expression of BBX24

Then, we evaluated the expression of *BBX24* using the information available on the *ePlant* website (http://bar.utoronto.ca/ Accessed on 3 September 2021). We found that *BBX24* expression is repressed in dry seeds and increased during the first 48 h of imbibition in water and light (Appendix A). To confirm these results in our conditions, we generated two independent transgenic lines of the *BBX24* promoter tagged with GUS. We sowed the seeds of p*BBX24*::GUS (lines 31 and 38) in water. The stratified and imbibed seeds were exposed to white light and dark cycles (16/8 h, respectively) during the first 30 h. We found that the GUS signal was low during the first 24 h, then increased significantly between 24 and 26 h and decreased slightly after 28 h (Figure 2A). The GUS staining appeared in the tip of the radicle of the embryo and then expanded to the adjacent tissues. Then, we evaluated the GUS signal in seeds of both lines, imbibed in moderated stresses of NaCl and PEG at 26 h and 30 h after light exposure. As expected, the GUS signal was strong in seeds imbibed in water, increasing significantly between 26 h and 30 h (Figure 2B). In contrast, the GUS staining in the radicle of seeds imbibed in NaCl and PEG was very low and similar between 26 h and 30 h, suggesting that *BBX24* expression is strongly repressed by both stresses (Figure 2B).

### 2.3. ABA Inhibits Seed Germination Promoted by BBX24

Salt and drought stresses are well known to be mediated by ABA in plants [2,3]. Therefore, we evaluated the effect of ABA on germinating Col-0, *bbx24-1* and *BBX24ox* seeds imbibed in water or 3 µM ABA. In water, the germination was 100% independent of the genotype at the end of the experiment (Figure 3A). In ABA, *BBX24ox* seeds germinated at a higher rate than Col-0 seeds during the first 72 h after the beginning of light exposure, but similar germination was documented at the end of the experiment. In contrast, *bbx24-1* seeds germinated at a significantly lower rate than Col-0 seeds at the end of the experiment. While Col-0 and *BBX24ox* seeds germinated at a rate higher than 60%, *bbx24-1* seeds germinated at ~30% (Figure 3A). A GUS staining experiment with two independent lines expressing p*BBX24*::GUS showed that the GUS signal was absent between 24 h and 30 h after the beginning of photoperiod, confirming that ABA represses the expression of *BBX24* in our conditions (Figure 3B). In addition, we designed a new experiment imbibing the p*BBX24*::GUS seeds of both lines with paclobutrazol (PAC), an inhibitor of GA. Previously, we performed an experiment to select the concentration of PAC (Appendix A). As expected, the reduction of GA synthesis by PAC inhibited seed germination and reduced p*BBX24*::GUS expression between 26 and 30 h of photoperiod (Appendix A). In addition, we evaluated the effect of 50 µM and 100 µM PAC on seed germination of Col-0 and *bbx24-1* seeds. In PAC, the germination was inhibited more strongly in *bbx24-1* than in Col-0 seeds. While Col-0 seeds germinated at ~50 and ~30% in 50 µM and 100 µM PAC, respectively, the *bbx24-1* seeds germinated at ~10% in both PAC treatments (Appendix A). Together, these results suggest that ABA and the reduction of GA in seeds inhibit the activity of the promoter of *BBX24* and, consequently, reduce the germination of seeds.

### 2.4. BBX24 Expression Is Induced by Light and ABA Inhibits Its Expression at Night

Since *BBX24* expression is light modulated in Arabidopsis plants [18,30], we studied *BBX24* expression in imbibed seeds during the first 16 h light/8 h dark photoperiod. In water, *BBX24* expression doubled after 6 h of exposition to light and then decreased slightly at 12 h. At the end of the dark period (i.e., 24 h), *BBX24* expression was reduced to the basal level (Figure 4). *BBX24* expression in seeds imbibed in ABA showed a similar pattern as in water, but with higher inhibition at the end of the night (Figure 4). These results suggest an inhibitory effect of ABA on *BBX24* expression, the effects of which are particularly significant at the end of the night.

### 2.5. BBX24 Reduces ABA Sensitivity and Increases NaCl Tolerance

ABA can exert its effects on the seeds by increasing ABA concentration or ABA sensitivity in seed tissues [7]. To distinguish between both effects, we designed a new experiment sowing the seeds into a medium with 100 µM fluoridone (Flu), an inhibitor of ABA synthesis, together with different doses of ABA. We found that increasing levels of ABA between 0 and 10 µM reduced the germination from 100% to ~10% in Col-0 and *bbx24-1* seeds (Figure 5A). ABA sensitivity was significantly lower between 3 and 5 µM of ABA in *bbx24-1* compared with Col-0 seeds (Figure 5A). These results suggest that BBX24 reduces the sensitivity of ABA. We also evaluated the effects of 200 mM NaCl on germination when seeds are imbibed at 0, 100 and 150 µM of Flu. Without Flu, Col-0 seeds germinated better than *bbx24-1* seeds. However, *bbx24-1* seeds germinated at the same levels of Col-0 seeds with 100 or 150 µM of Flu in a medium with NaCl (Figure 5B), suggesting that ABA synthesis is a prerequisite for the activity of BBX24.

### 2.6. BBX24 Increases GA Sensitivity in Imbibed Seeds in ABA

Knowing the relevance of the GA/ABA ratio for seed germination [7], we designed an experiment to study the effects of increasing levels of GA in Col-0 and *bbx24-1* seeds imbibed in 3 µM ABA. As expected, in water, Col-0 and *bbx24-1* seeds germinated completely (i.e., 99.5 ± 0.5%). In ABA, Col-0 seeds germinated at ~30%, slightly higher than *bbx24-1* seeds, but not significantly (Figure 6). Increasing levels of GA, between 3 and 50 µM, promoted ~80% germination of Col-0 seeds but did not produce any promotive effects in *bbx24-1* seeds (Figure 6). These results demonstrate that the activity of BBX24 on seed germination in ABA is affected by increasing levels of GA, suggesting that BBX24 promotes seed germination in ABA through the GA/ABA balance.

### 2.7. BBX24 Inhibits ABI5 y ABI3 in ABA Imbibed Seeds

To have a better understanding of the effect of BBX24 on ABA signaling, we evaluated the expression of genes previously known to be involved in seed germination. Imbibed seeds of Col-0 and *bbx24-1* were harvested at 12 h of the first photoperiod. RNA samples were analyzed to evaluate the expression of *ABI5*, *ABI3*, *XERICO* and *PIF6* (Figure 7). Gene expression was analyzed in seeds imbibed in water and 3 µM ABA. The expression of the four genes was not affected by ABA in Col-0 seeds. However, the expression of *ABI5* and *ABI3* increased significantly in *bbx24-1* seeds (Figure 7), suggesting that BBX24 can inhibit the expression of *ABI* genes contributing to the promotive effects on germination under moderated stress.

### 2.8. BBX24 Is Genetically Epistatic to ABI5 and HY5

Regarding the previous results, we evaluated the genetic interaction between *BBX24*, *HY5* and *ABI5*, known components in ABA signaling [11,12,31]. Using genetic crosses between *bbx24-1*, *abi5-8* and *hy5-215* mutants, we selected double and triple mutants to study the genetic relationship between those genes. In water, the germination was 100% in Col-0, *bbx24-1*, *bbx24-1 abi5-8*, *bbx24-1 hy5-215* and *bbx24-1 abi5 hy5-215* seeds (Figure 8, Appendix A. In 5 µM of ABA, Col-0 seeds germinated at a higher rate than *bbx24-1* seeds (i.e., ~50% vs. ~25%, respectively). Double and triple mutant seeds germinated quicker than Col-0 seeds, reaching 100% of germination at 96 h (Figure 8, Appendix A. These results suggest that BBX24 is genetically epistatic to *ABI5* and *HY5* and acts upstream of these genes.

## 3. Discussion

Overall, the results of this work demonstrate that the activity of BBX24 increases osmotic and salt tolerance in germinating Arabidopsis seeds. Previous studies found that the overexpression of BBX24 can increase osmotic or saline tolerance in adult plants. For example, the overexpression of STO/BBX24 in transgenic Arabidopsis plants produces higher salt tolerance increasing root growth [27]. In another study, Yang et al. (2014) found a decreased tolerance to freezing and drought stresses in Cm-BBX24 silencing lines of chrysanthemum. BBX24-silenced lines of *Solanum tuberosum* plants have also been documented to have a lower saline tolerance [32]. BBX24-silenced potato plants watered with 0.15 M of NaCl for 8 days produced lower chlorophyll content (~40%) and biomass compared with the wild-type control. However, the overexpression of BBX24 did not produce higher levels of chlorophylls than wild-type suggesting that endogenous levels of BBX24 saturate this response under NaCl stress [32]. Then, the results in adult plants of different species agree with the germination results of our work exposing *bbx24-1* seeds under osmotic and saline stresses.

Germination is finely regulated by the GA/ABA balance in seeds. We found that *bbx24-1* mutant seeds germinate at a lower rate than Col-0 seeds in ABA, and the expression of BBX24 is significantly lower in ABA with respect to control seeds imbibed in water at the end of the night (Figure 3 and Figure 4). Interestingly, two additional lines of evidence suggest that BBX24 increases the tolerance to ABA because *bbx24-1* seeds are more sensitive to exogenous ABA when the synthesis of endogenous ABA is inhibited by Flu, and they are also more tolerant to NaCl at higher doses of Flu (Figure 5). Apparently, the function of BBX24 is to increase the GA sensitivity in seeds because *bbx24-1* seeds are insensitive to the addition of GA_4_ in the medium of imbibition (Figure 6). Our results suggest that BBX24 is genetically epistatic to ABI5 and HY5 and acts upstream of these genes controlling the expression of *ABI3* and *ABI5* when seeds are imbibed in a solution with ABA (Figure 7 and Figure 8). The negative interaction between BBX24 and HY5, and its closest homolog HYH, is a common theme in the regulation of seedling photomorphogenesis [17,18,19,22]. In simulated shade, BBX24 can sequester DELLA to promote elongation responses through PIF4 activity [23,24]. More recently, it has been suggested that the module of HY5-BBX24-DELLA can control the crosstalk between GA and UV-B signaling to regulate hypocotyl inhibition [21]. Together, our results suggest that BBX24 can promote seed germination in suboptimal stressful environments by integrating ABA and GA signals.

BBX24 is induced in the radicle tips of seeds during the first hours after the beginning of the light treatment, and PEG, NaCl and ABA inhibit its activity (Figure 2 and Figure 3). The *BBX24* gene expression increased during the first 6 h of exposition of the seed to light and then decreased to basal levels during the night, the inhibitory effects being higher when seeds are imbibed in a solution with ABA (Figure 4). The induction of *BBX24* expression by light and the circadian clock has been previously documented in Arabidopsis [18,30] and *Solanum* spp. [33]. The expression of *BBX24* was induced in five-day-old seedlings exposed to continuous light and also in six-week-old plants cultivated in short-day conditions in Arabidopsis [18]. Similar results were documented in two-week-old potato plants cultivated in a photoperiod of 14 h/10 h light/dark cycles [33]. As we documented here in Arabidopsis seeds, the expression of *BBX24* decreased when potato plants were treated with drought or irrigated with saline solutions [33]. As the circadian clock is involved in the integration of alternating temperatures and light in Arabidopsis seeds [34], we speculate that BBX24 could contribute to the synchronization of germination in the window time when conditions are more favorable for the successful establishment of the future seedlings.

Additional research is needed to have a more comprehensive understanding of the molecular mechanisms of BBX24 on seed germination. Identifying the transcription factors and elucidating the mechanisms involved in the adjustment of germinating seeds in sub-optimal conditions can help to generate more tolerant genotypes to ensure sustainable agriculture in climate change contexts.

## 4. Materials and Methods

### 4.1. Plant Material and Plant Cultivation

The experiments were carried out using *A. thaliana* seeds of Col-0, BBX24 overexpressing line (*BBX24ox*) [23]; *bbx24-1* (AT1G06040) [17], *abi5-8* (AT2G36270, SALK_013163C), *hy5-215* (AT5G11260) [35]. The seeds were obtained from plants grown together under the same cultivation conditions. Seeds were sown in transparent acrylic boxes 4 × 4 cm with agar-water 0.8% (p/v) and stratified at 5 °C in the dark for 4 days. The boxes were exposed to white light in a long-day chamber (16 h light/8 h dark) at 22 °C to induce germination. Seven-day-old seedlings were transplanted into 250 cm^3^ plastic pots of 8 cm diameter with a mixture of vermiculite, perlite and peat (2:1:1). Irrigation was carried out regularly with Hakaphos Red 1 g/L fertilizer. Seeds were harvested and stored in envelopes in a box with silica gel in darkness at 5 °C until further use. Detailed experimental protocols are shown in Appendix A.

### 4.2. Selection of Mutants and Generation of pBBX24::GUS Lines

New lines of mutants were generated by single crossing between *bbx24-1* and *abi5-8* or *hy5-215*. The new materials were selected by genotyping the second generation of plants (F2) via PCR using the primers indicated in Appendix A. Fresh leaf tissue was collected and DNA extracted using the CTAB-based method.

For the construction of p*BBX24*::GUS lines, 1541 base pairs upstream of the start codon of the promoter of BBX24 gene were amplified with the primers FW: 5′-GGGGACAAGTTTGTACAAAAAAGCAGGCTTGTGCAAACATTGTCAAAGGC-3′ and RV: 5′-OkGGGGACCACTTTGTACAAGAAAGCTGGGTTCACCGGATACAAGAAACAAAATATC-3′. The promoter was cloned into the pZeo donor vector via a recombination reaction using Gateway™ BP Clonase™ (ThermoFisher Scientific, Waltham, MA, USA), following the manufacturer’s instructions. The BBX24 gene promoter was subcloned into the destination vector pBGWFS7 via a recombination reaction with the enzyme Gateway™ LR Clonase™ (ThermoFisher Scientific), following the manufacturer’s instructions. The construction pBGWFS7·pBBX24 contains the sequence of the GUS and GFP reporter genes under the promoter of the BBX24 gene.

To obtain transgenic plants, transformation with *Agrobacterium tumefaciens* was performed using the floral dip method [36]. The *A. tumefaciens* cells were transformed with the plasmid of interest and grown in liquid culture with the corresponding antibiotics at 28 °C. Then, the culture was precipitated in an ultracentrifuge for 20 min at 4000× *g* and resuspended in a 5% and 0.01% sucrose solution of Silwet that was used to immerse flowers of four-week-old Arabidopsis plants. Seeds from these plants (T0) were harvested and sorted on plates with MS medium supplemented with 10 mg/L BASTA. Seeds from these plants (T1) were sown in a selective medium and only those lines with a 3:1 resistance ratio were selected. The offspring of these plants (T2) were sown in a selective medium and lines from homozygous plants were selected with a resistance ratio of 4:4 (lines 31 and 38). The experiments were performed with homozygous T3 or T4 lines.

### 4.3. Seed Germination Experiments

The seeds were sown on filter paper with different media placed in plastic boxes of size 4 × 4 cm, maintaining a control treatment in water. The boxes with seeds were incubated at 5 °C for 3 days to homogenize germination. Then, the boxes were placed in a white light chamber with a long-day photoperiod (16 h light/8 h dark) at 22 °C for 7 days. Seed germination was counted daily. The criterion for germination was the emergence of the radicle evaluated with a stereoscopic magnifying lens (Luxeo 4D, LABOMED, Mendoza, Argentina.

For the experiments under stress, we did a calibration curve with Col-0 seeds using different doses of NaCl (BIOPACK) and PEG6000 (Fluka, Sigma-Aldrich, Darmstadt, Germany). The seeds were sown with increasing concentrations of NaCl = 0, 50, 75, 100, 150, 200 and 250 mM and PEG6000 with Ψa = 0; −0.2; −0.3; −0.4; −0.6 and −0.8 MPa. We used moderated concentrations of NaCl = 150 mM and PEG = −0.4 Mpa, which inhibited the germination of Col-0 seeds by ~50%.

For the experiments with hormones, we did a calibration curve with Col-0 seeds using doses of 0, 1, 3 and 5 µM ABA (Sigma-Aldrich) or 0, 5, 10, 50, 100, 150 and 200 µM Paclobutrazol (PAC, Sigma-Aldrich), an inhibitor of GA. We used moderated concentrations of ABA = 3 µM and PAC = 50 µM, which inhibited the germination of Col-0 seeds by ~50%.

We designed two additional experiments using 150 mM of NaCl with increasing doses of GA_4_ (Sigma-Aldrich) or fluridone (Flu; Sigma-Aldrich), an inhibitor of ABA synthesis, to evaluate the combined effects of salt stress and hormones.

### 4.4. B-Glucuronidase Staining Assay

Seeds of two independent lines, p*BBX24*::GUS (31 and 38), were sown in different media to evaluate the expression and localization of BBX24. Samples were harvested at indicated imbibition times and fixed in 90% acetone for 20 min, followed by two successive washes with sterile water. Then, they were incubated in X-Gluc solution (100 mM phosphate buffer pH 7.0; 0.8 mM X-Gluc; 10% triton; 100 mM Ferro; 100 mM Ferri) at 37 °C until staining was observed in the tissues with a stereoscopic magnifying lens (Luxeo 4D, LABOMED). The seeds were preserved in 70% alcohol, at −20 °C until the photographs were taken when the embryos were separated from their covers using forceps and a surgical needle. GUS expression was quantified in embryo images as described previously [37]. Briefly, the images were converted to HSB type, and the channel “saturation” was selected utilizing the software ImageJ. An increase in saturation depicts a purer color, which reflects the intensity of the GUS dye present in the sample. The region of interest (ROI) was selected in a rectangular shape, and the intensity values of the selected area were measured. At the same time, we evaluated the germination response of the seeds.

### 4.5. q-PCR Expression Analysis

RNA was extracted from seeds (5–10 mg per replicate) using an RNeasy Plant Mini Kit (Qiagen), following the manufacturer’s protocol. To obtain cDNA, 2 µg of total RNA was mixed with 2 µL of oligo-dT primer and incubated at 65 °C for 5 min. It was immediately brought to 4 °C. Then, 4 µL of 5X buffer, 1 µL of dNTPs, 250 µM of each (1:1:1:1), 2 µL of 100 mM DTT, 1 µL of RNA nuclease inhibitor (20 U/µL) and 0.2 µL reverse transcriptase (SuperScript IIRT, Invitrogen, Waltham, MA, USA) (200 U/µL) were added. Amplification was carried out with FastStart Universal SYBR Green Master, Rox (Roche, Bassel, Switzerland), using 2 µL of the cDNA dilution per reaction. The PCR reaction and fluorescence quantification were performed with an ABI 7500 Real-Time PCR System (Applied Biosystems, Waltham, MA, USA). Three independent samples were measured for each point. Two technical replicates were made of each sample and the average of these two values was taken as a repetition. The expression data of the genes of each sample were relativized to the level of expression of the UBC gene [38]. The primers used are detailed in Appendix A.

### 4.6. Statistical Analysis

Statistical analysis was performed using two-way analysis of variance (ANOVA) with genotype and treatments as factors. Each replicate consisted of 30 seeds (otherwise indicated in the text). The germination experiments were repeated in time. The means of the replicates were compared with Col-0 using the Bonferroni comparison test. Graphics and statistical tests were performed with GraphPad PRISMA v9 (https://www.graphpad.com/ access on 1 February 2021).

## Figures and Tables

**Figure 1 plants-12-02392-f001:**
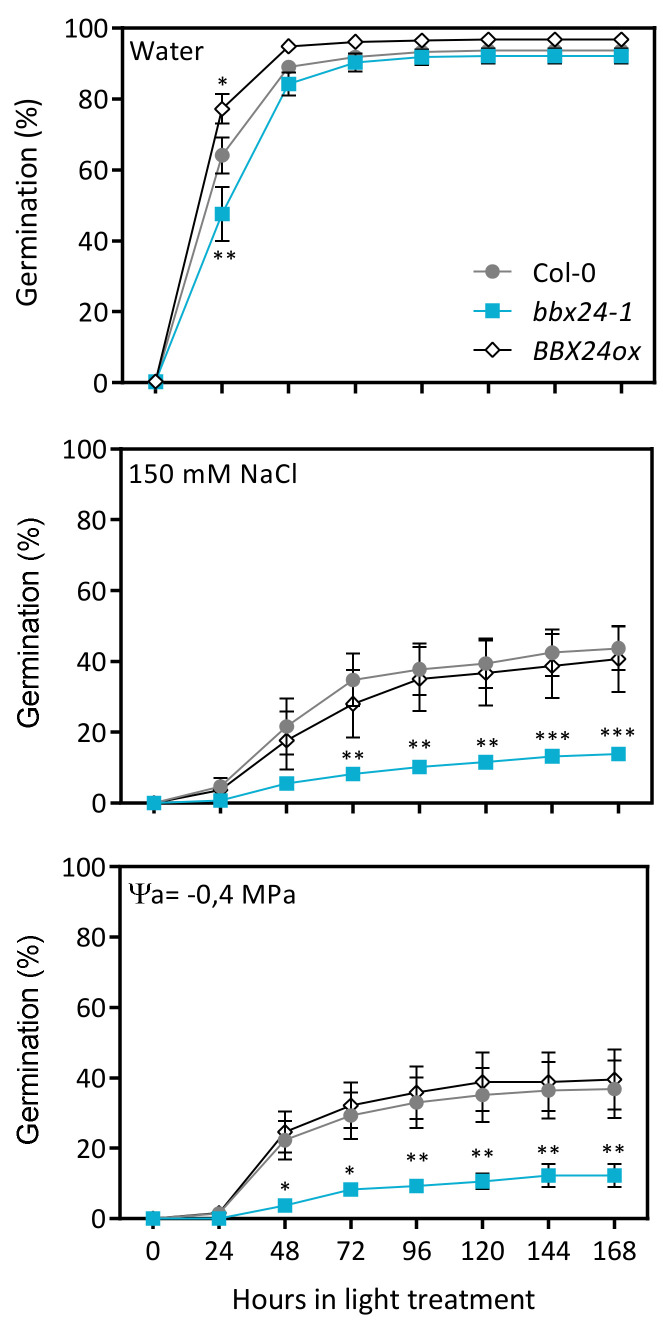
BBX24 promotes seed germination in drought and NaCl. Seeds of Col-0, *bbx24-1* and *BBX24ox* were sown in water, 150 mM NaCl and PEG 6000 (Ψa = −0.4 MPa), stratified for 3 days at 5 °C in darkness, and then transferred to 16 h/8 h light/darkness at 22 °C until germination was assessed. Each point indicates mean ± SEM (*n* = 12). Comparisons between means for *bbx24-1* or *BBX24ox* with Col-0 at the same condition are indicated via the Bonferroni test (* *p* < 0.05, ** *p* < 0.01 and *** *p* < 0.001).

**Figure 2 plants-12-02392-f002:**
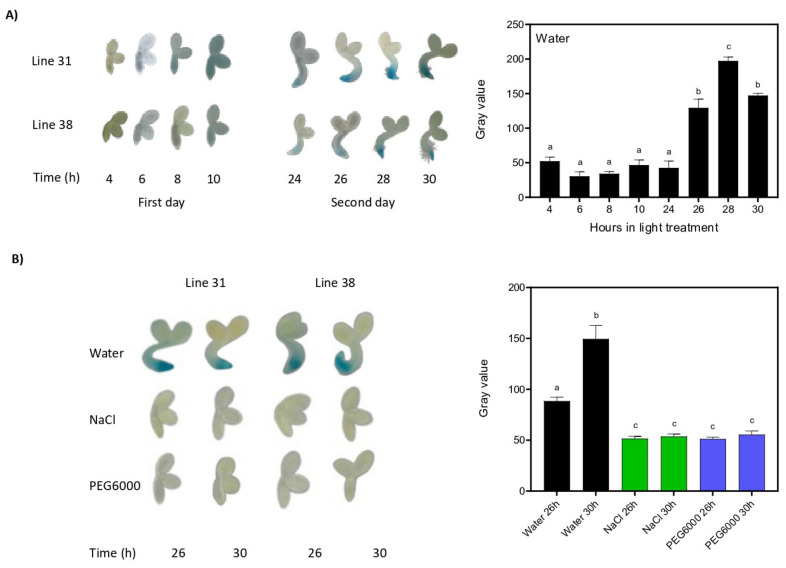
PEG and NaCl inhibit the expression of BBX24. (**A**) Representative images show the expression of *BBX24* in embryos of two independent transgenic lines carrying p*BBX24::GUS* (31 and 38), taken at different time points between 4 h and 30 h after light exposure. The blue color indicates GUS expression. On the right, the quantification of histochemical staining of β-glucuronidase was performed with the software ImageJ. Gray values depicting GUS activity were quantified in embryos between 4 h and 30 h after light exposure. (**B**) The photographs show the expression of *BBX24* in embryos of two independent transgenic lines expressing p*BBX24::GUS* embryos imbibed in water, 150 mM NaCl o PEG 6000 (Ψa = −0.4 MPa) at 26 h or 30 h after light exposure. For other references, see (**A**). Each bar indicates mean ± SEM. Different letters indicate significant differences between means obtained via two-way ANOVA followed by the Bonferroni test (*p* < 0.05).

**Figure 3 plants-12-02392-f003:**
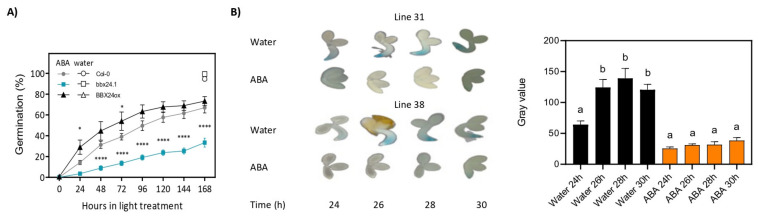
ABA inhibits seed germination promoted by BBX24. (**A**) Seeds of Col-0, *bbx24-1* and *BBX24ox* were sown in water or 3 µM ABA, stratified for 3 days at 5 °C in darkness, and then transferred to 16 h/8 h light/darkness at 22 °C until germination was counted. Germination in water at the end of the experiment is indicated with open symbols. Each point indicates mean ± SEM (n = 6). Comparisons between means with Col-0 at the same conditions were done via the Bonferroni test (* *p* < 0.05 and **** *p* < 0.0001). (**B**) Representative images showing the expression of *BBX24* in embryos of two independent transgenic lines p*BBX24::GUS* at 24, 26, 28 and 30 h after light exposure. The seeds were sown in water or 3 µM ABA, stratified for 3 days at 5 °C in darkness and then transferred to 16 h/8 h light/darkness at 22 °C until GUS expression was assessed via histochemical staining. Each point indicates mean ± SEM (n = 3). Different letters indicate significantly different means obtained via two-way ANOVA followed by the Bonferroni test (*p* < 0.05).

**Figure 4 plants-12-02392-f004:**
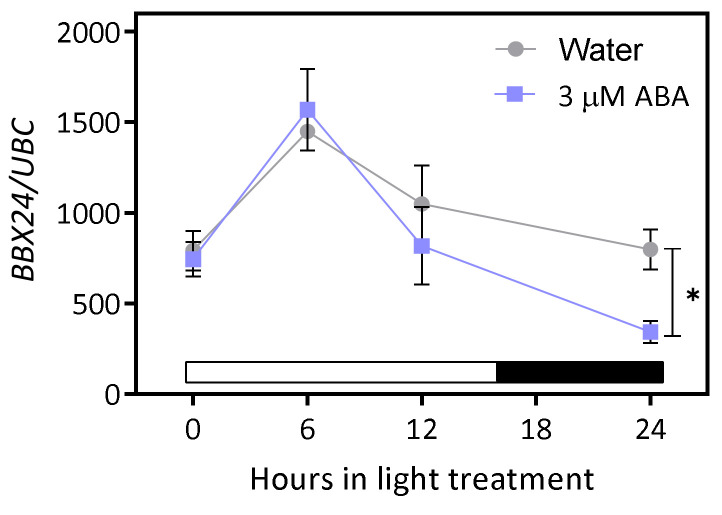
BBX24 is induced by light and ABA inhibits its expression at the end of the night. *BBX24* expression via qPCR in Col-0 seeds imbibed in water or ABA 3 µM. The seeds were sown, stratified for 3 days at 5 °C in darkness, and then transferred to 16 h/8 h light/darkness at 22 °C. Expression was quantified during the first photoperiod at 0, 6, 12 and 24 h. Each point indicates mean ± SEM (n = 3). Significant differences between means at the same time are indicated via the *t* Student test (* *p* < 0.05).

**Figure 5 plants-12-02392-f005:**
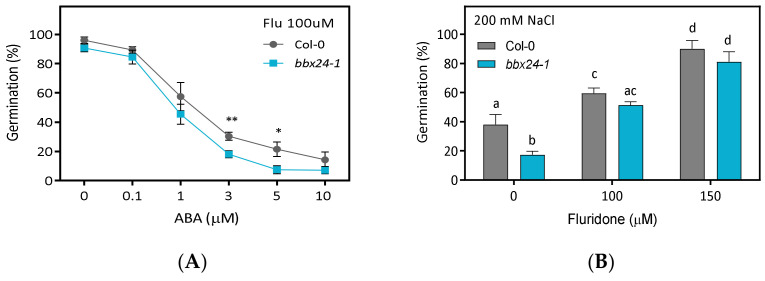
BBX24 reduces ABA sensitivity and increases NaCl tolerance. Seeds of Col-0 and *bbx24-1* were sown in (**A**) 100 µM of fluridone (Flu) with the addition of different doses of ABA or (**B**) in 200 µM NaCl with the addition of 0, 100 or 150 µM Flu. The seeds were stratified for 3 days at 5 °C in darkness and then transferred to 16 h/8 h light/darkness at 22 °C until germination was counted at the end of the experiment. Each point indicates mean ± SEM (n = 8). Comparisons between means were done via the Bonferroni test. In (**A**), significant differences with Col-0 at the same condition are indicated by * *p* < 0.05, ** *p* < 0.01, and in (**B**), different letters indicate significantly different means obtained via two-way-ANOVA followed by the Bonferroni test (*p* < 0.05).

**Figure 6 plants-12-02392-f006:**
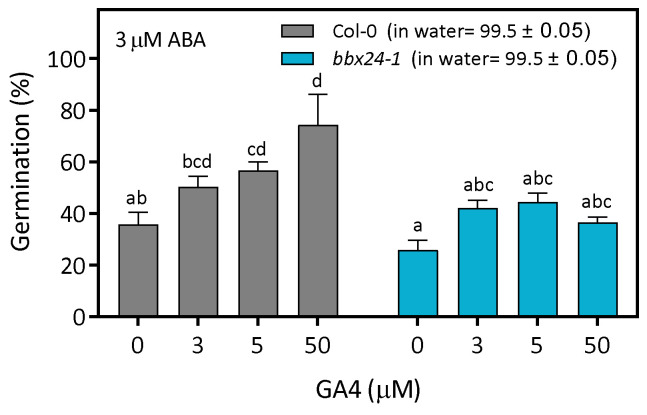
BBX24 increases GA sensitivity in imbibed seeds in ABA. Seeds of Col-0 and *bbx24-1* were sown in 3 µM of ABA with the addition of different doses of GA4. The seeds were stratified for 3 days at 5 °C in darkness and then transferred to 16 h/8 h light/darkness at 22 °C until the germination was counted at the end of the experiment. Each point indicates mean ± SEM (n = 6). Comparisons between means were done via the Bonferroni test. Different letters indicate significant differences between means (*p* < 0.05).

**Figure 7 plants-12-02392-f007:**
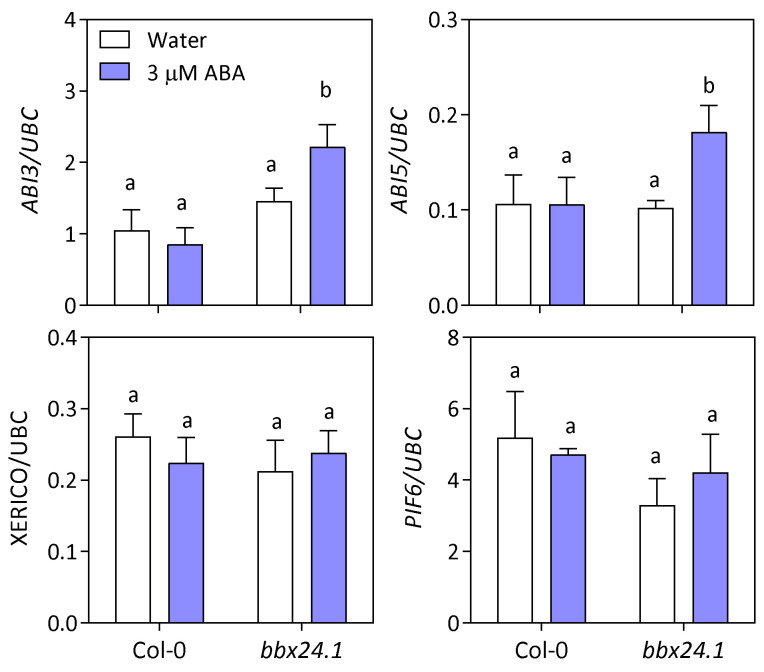
BBX24 inhibits *ABI5* and *ABI3* in seeds imbibed in ABA. *ABI3*, *ABI5*, *XERICO* and *PIF6* expression via qPCR in Col-0 or *bbx24-1* seeds, imbibed in water or ABA 3 µM. The seeds were sown, stratified for 3 d at 5 °C in darkness, and then transferred to 16 h/8 h light/darkness at 22 °C. The expression was quantified at 12 h of the first photoperiod. Each bar indicates mean ± SEM (n = 3). Comparisons between means were done via the Bonferroni test. Different letters indicate significant differences between means (*p* < 0.05).

**Figure 8 plants-12-02392-f008:**
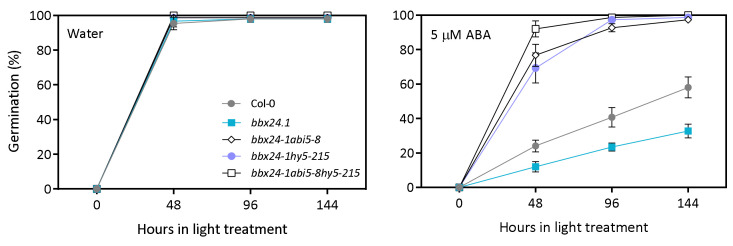
BBX24 is genetically epistatic to ABI5 and HY5. Seeds of Col-0, *bbx24-1*, *bbx24-1 hy5-215*, *bbx24-1 abi5-8* and *bx24-1 hy5-215 abi5-8* were sown in water or 5 µM ABA, stratified for 3 d at 5 °C in darkness, and then transferred to 16 h/8 h light/darkness at 22 °C until germination was counted. Each point indicates mean ± SEM (n = 3). Each replicate consisted of 50 seeds.

## Data Availability

The authors confirm that the data supporting the findings of this study are available within the article and its Appendix A.

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
