# Peer review of "BBX24 Increases Saline and Osmotic Tolerance through ABA Signaling in Arabidopsis Seeds"

_plants, 2023, doi:10.3390/plants12132392_

Round 1

Reviewer 1 Report

Thank you for considering me to revise the manuscript titled “BBX24 promotes seed germination through ABA signaling under saline and osmotic stresses.”. The manuscript provides good information on the role of BBX24 in enhancing seed germination under abiotic stress (drought and salinity). Indeed, the authors have put a great effort into this study. Generally, the manuscript is well-written and well structured accordingly it could be accepted after minor revision. 

Specific comments

The manuscript needs major English editing; long and less informative sentences, spelling and grammatical errors should be carefully revised throughout the manuscript.

The title needs to be improved, could be modified to be “Assessing the cellular and molecular role of BBX24 on seed germination under different abiotic stresses”

The methodology is poorly presented in the abstract, needs to be improved with more details.

More information on BBX family should be added to the introduction, the importance of BBX24 on seed germination, knowledge gap, rationale, hypothesis, and objectives need more clarification.

Line 198 “Experimental design” should be "statistical analysis” and the experimental design of the applied trials should be presented earlier in Ms&MS.

In Figures 6-7, the significance letter “a” should correspond with the highest value while "d" with the lowest value.

The obtained results are poorly discussed, I suggest presenting the discussion in a separate section and should be improved. The reference list contains 36 references, of which about 25 are presented in the introduction. More updated references should be considered in the discussion section. 

The authors cited previously published reports in the conclusion which is not preferred. The actual conclusion could be integrated into the discussion section and the conclusion should be rewritten. 

The reference style (in the text and the reference list) should follow the journal instructions. The reference should be carefully revised in the text and reference list. Sometimes, the journal name is abbreviated (as Plant J in line 501, Curr Opin Plant Biol line 528, Front Plant Sci line 532) while most journals are not abbreviated.  The reference list should be revised and modified.

The manuscript needs major English editing; long and less informative sentences, spelling and grammatical errors should be carefully revised throughout the manuscript.

Author Response

Thank you for considering me to revise the manuscript titled “BBX24 promotes seed germination through ABA signaling under saline and osmotic stresses.”. The manuscript provides good information on the role of BBX24 in enhancing seed germination under abiotic stress (drought and salinity). Indeed, the authors have put a great effort into this study. Generally, the manuscript is well-written and well structured accordingly it could be accepted after minor revision. 

Authors are grateful for your interest in our manuscript and for your kind comments. We agree with the reviewer that our report is a nice piece of scientific work.

Specific comments

The manuscript needs major English editing; long and less informative sentences, spelling and grammatical errors should be carefully revised throughout the manuscript.

The English was edited and revised.

The title needs to be improved, could be modified to be “Assessing the cellular and molecular role of BBX24 on seed germination under different abiotic stresses”

We agree with the reviewer that BBX24 is related with cellular and molecular events on seed germination, but we consider that the results of our work demonstrate the critical role of BBX24 to increase stress tolerance in seeds through ABA. We improved the title of the work as suggested by the reviewer and now reads “BBX24 increases saline and osmotic tolerance through ABA signaling”.

The methodology is poorly presented in the abstract, needs to be improved with more details.

We included a sentence with more methodological details in the abstract.

More information on BBX family should be added to the introduction, the importance of BBX24 on seed germination, knowledge gap, rationale, hypothesis, and objectives need more clarification.

Thank you for this comment and now the introduction reads “BBX24 has been documented to inhibit seedling de-etiolation (Indorf et al 2007, Yan et al 2011, Gangappa et al 2013), repress UVB signaling (Jiang et al 2012), and promote shade avoidance (Gangappa et al 2013, Crocco et al 2015) and flowering (Li et al 2014). BBX24 was originally isolated as a SALT-TOLERANT (STO) protein in a Arabidopsis cDNA clone screen that confer increased salt-tolerance in Saccharomyces cerevisiae yeast (Lipunner et al 1996; Nagaoka et al 2003) and the BBX24 overexpression increases salt tolerance in Arabidopsis (Nagaoka et al 2003).” In addition, the first sentence of the last paragraph reads “However, the function of BBX24 in seed germination is completely unknow. Regarding, the relevance of BBX24 increasing salt tolerance (Lipunner et al 1996; Nagaoka et al 2003), this work aimed to study the function of BBX24 in the control of germination during the imbibition of seeds in saline and osmotic conditions.”

Line 198 “Experimental design” should be "statistical analysis” and the experimental design of the applied trials should be presented earlier in Ms&MS.

It was done.

In Figures 6-7, the significance letter “a” should correspond with the highest value while "d" with the lowest value.

We used the same style in all figures to indicate significance differences between means by letter in the same order. If the editorial office requires to change the style, we can do it.

The obtained results are poorly discussed, I suggest presenting the discussion in a separate section and should be improved. The reference list contains 36 references, of which about 25 are presented in the introduction. More updated references should be considered in the discussion section. The authors cited previously published reports in the conclusion which is not preferred. The actual conclusion could be integrated into the discussion section and the conclusion should be rewritten. 

Thank you very much for this suggestion. We edited the discussion as an independent section and the work improved.

The reference style (in the text and the reference list) should follow the journal instructions. The reference should be carefully revised in the text and reference list. Sometimes, the journal name is abbreviated (as Plant J in line 501, Curr Opin Plant Biol line 528, Front Plant Sci line 532) while most journals are not abbreviated.  The reference list should be revised and modified.

It was done.

Comments on the Quality of English Language

The manuscript needs major English editing; long and less informative sentences, spelling and grammatical errors should be carefully revised throughout the manuscript.

The English was revised.

Reviewer 2 Report

The work of Chiriotto et al., is a very interesting study and deserve to be published in PLANTS.

A significant volume of data have been produced and presented in this manuscript which in my view need a much better organization and arrangement. The writing is suffering from inconsistency and a lack of unified orientation. I do suggest to the authors to reformulate the sections and clear up the text from unnecessary details and instead focus on their findings and provide a deeper discussion of the results with already published works. Even a separate discussion section can be dedicated to this work which has a lot of data produced.

Here are some minor comments and suggestion that maybe of some use to improve their contribution:

Line 37-40: Please rewrite the sentence to not to be the same as line 13-15 of the abstract.

Line 46: you can add another reference instead of the ’’Bewley 2013’’ that was referred to earlier in the text. doi: 10.3390/agriculture10040094

Line 79: Is ‘’Pharmacological experiments’’ a right? Bring a reference after such statements.

Line 80-81: ‘’BBX24 also increases GA sensitivity when seeds are imbibed in ABA solution’’ Bring a reference after such statements.

Line 81-83: ‘’At the molecular level, BBX24 inhibits the expression of ABI3 and ABI5 in ABA, and BBX24 genetically interacts upstream of HY5and ABI5’’. Bring a reference.

Line 83: change ‘’demonstrates’’ to ‘investigates’.

Line 94: Give a better description for ‘’250 cm3 plastic cups’’

Line 99: ‘’genotyped’’ or ‘genotyping’?

Line 100: ‘’Freshly’’ or ‘Fresh’?

Line: 86: I suggest to keep the content of this subsection (2.1) relevant to its title or add the DNA extraction and PCR to its title.

Line 100-110: remove the details of the DNA extraction. It is enough to just mention a CTAB-Based method and the obtained DNA quantity range.

Line 113-115: always put 5’ and 3’ before and at the end of the primers sequences. The same in the Supl. Table 1.

Line 116: give the origin/vendor of the applied vector.

Line 130: how did you detect the ‘’as expected segregation for lines with a single copy of T-DNA insertion in the genome’’. ?

Line 150, 151, 155, 167 and elsewhere: remove the https://www.sigmaaldrich.com/ or such links.

Line 163: use a unified version of ºC or °C throughout the text

Line: 175: I’m wondering how you could get 2µg of RNA from 5-10 mg of plant samples specially using RneasyPlant Mini Kit and why did you diluted 1 µl of sample in 200 µl of H2Omq (MQ H2O)

Line 177-186. You do not need to give the kit’s manual details, just name the applied kit and the RT enzyme name. But considering the numbers and volumes and concentrations, double check this section. Finally you have 20 µl of cDNA that was already diluted a few times. What about the normalization of the RNA samples quantity?

Line 198: change ’’Experimental design’’ this to ‘Statistical analyses, and correct the terminology.

‘’…experimental design was an ANOVA design’’ is technically incorrect. ANOVA is referred to the analysis of the data which in even in your case it seems to me that you used MANOVA if you used Bonferroni correction! Please check this with a statistician and correct the terminology. For the experimental design, I guess you have used the CRD!

 Line 247: ‘’exposed to white light and dark’’. Such details are missing from the methods section. Please comprehensively reformulate the methods and include more details about the conducted treatments. I suggest showing a schematic graphic work or illustration to visualizes the treatment/sampling/times/analysis etc.

Line 330: ‘’…new experiment sowing the seeds into a medium with 100 µM fluoridone (Flu)’’. Such details are missing from the methods section. Please comprehensively reformulate the methods and include more details about the conducted treatments. It is just confusing to suddenly bring a new experiment in to the text as it was not reported in the methods section.

 Line 379: Fig 7. Better adjust the letter’s box to not cover thee error bars.

Supplementary materials: Please give caption for all figures and tables. Give the PCR products sizes.

Supplementary Fig 3 is not mentioned in the text.

Should be improved

Author Response

The work of Chiriotto et al., is a very interesting study and deserve to be published in PLANTS.

A significant volume of data have been produced and presented in this manuscript which in my view need a much better organization and arrangement. The writing is suffering from inconsistency and a lack of unified orientation. I do suggest to the authors to reformulate the sections and clear up the text from unnecessary details and instead focus on their findings and provide a deeper discussion of the results with already published works. Even a separate discussion section can be dedicated to this work which has a lot of data produced.

Authors are grateful for your interest in our manuscript and for your kind comments. We separate the discussion as suggested for the reviewer.

Here are some minor comments and suggestion that maybe of some use to improve their contribution: 

Line 37-40: Please rewrite the sentence to not to be the same as line 13-15 of the abstract.

It was done.

Line 46: you can add another reference instead of the ’’Bewley 2013’’ that was referred to earlier in the text. doi: 10.3390/agriculture10040094 

It was done.

Line 79: Is ‘’Pharmacological experiments’’ a right? Bring a reference after such statements.

It was changed. The sentence refers to the results of our experiments.

Line 80-81: ‘’BBX24 also increases GA sensitivity when seeds are imbibed in ABA solution’’ Bring a reference after such statements.

The sentence refers to the results of our experiments.

Line 81-83: ‘’At the molecular level, BBX24 inhibits the expression of ABI3 and ABI5 in ABA, and BBX24 genetically interacts upstream of HY5and ABI5’’. Bring a reference.

The sentence refers to the results of our experiments.

Line 83: change ‘’demonstrates’’ to ‘investigates’.

It was done.

Line 94: Give a better description for ‘’250 cm3 plastic cups’’

It was done.

Line 99: ‘’genotyped’’ or ‘genotyping’?

It was done.

Line 100: ‘’Freshly’’ or ‘Fresh’?

It was done.

Line: 86: I suggest to keep the content of this subsection (2.1) relevant to its title or add the DNA extraction and PCR to its title.

We included the mutant selection method into section 2.2.

Line 100-110: remove the details of the DNA extraction. It is enough to just mention a CTAB-Based method and the obtained DNA quantity range.

It was done.

Line 113-115: always put 5’ and 3’ before and at the end of the primers sequences. The same in the Supl. Table 1.

It was done.

Line 116: give the origin/vendor of the applied vector.

It was included.

Line 130: how did you detect the ‘’as expected segregation for lines with a single copy of T-DNA insertion in the genome’’. ?

We corrected the mistake.

Line 150, 151, 155, 167 and elsewhere: remove the https://www.sigmaaldrich.com/ or such links.

It was done.

Line 163: use a unified version of ºC or °C throughout the text

It was done.

Line: 175: I’m wondering how you could get 2µg of RNA from 5-10 mg of plant samples specially using RneasyPlant Mini Kit and why did you diluted 1 µl of sample in 200 µl of H2Omq (MQ H2O)

We corrected the mistake.

Line 177-186. You do not need to give the kit’s manual details, just name the applied kit and the RT enzyme name. But considering the numbers and volumes and concentrations, double check this section. Finally you have 20 µl of cDNA that was already diluted a few times. What about the normalization of the RNA samples quantity?

It was done. The RNA samples were normalized to the expression of UBC reference gene. The text reads “The expression data of the genes of each sample were relativized to the level of expression of the UBC gene (Czechowski et al. 2005).”

Line 198: change ’’Experimental design’’ this to ‘Statistical analyses, and correct the terminology. ‘’…experimental design was an ANOVA design’’ is technically incorrect. ANOVA is referred to the analysis of the data which in even in your case it seems to me that you used MANOVA if you used Bonferroni correction! Please check this with a statistician and correct the terminology. For the experimental design, I guess you have used the CRD!

Thank you for this comment. The text now reads “Statistical analyses were performed using two-way analysis of variance (ANOVA) with genotype and treatments as factors.”

 Line 247: ‘’exposed to white light and dark’’. Such details are missing from the methods section. Please comprehensively reformulate the methods and include more details about the conducted treatments. I suggest showing a schematic graphic work or illustration to visualizes the treatment/sampling/times/analysis etc.

We included a diagram for the experimental protocols in the Suppl. Fig. 1

Line 330: ‘’…new experiment sowing the seeds into a medium with 100 µM fluoridone (Flu)’’. Such details are missing from the methods section. Please comprehensively reformulate the methods and include more details about the conducted treatments. It is just confusing to suddenly bring a new experiment in to the text as it was not reported in the methods section.

The experiment is included in the section 2.3. The text reads “We designed two additional experiments using 150 mM of NaCl with increasing doses of GA4 (Sigma-Aldrich) or fluridone (Flu; Sigma-Aldrich), an inhibitor of ABA synthesis, to evaluate the combined effects of salt stress and hormones. “

 Line 379: Fig 7. Better adjust the letter’s box to not cover thee error bars. 

Apparently, it was an error of pdf conversion. We checked all the figures for this resubmission and replaced those that appears in bad quality.

Supplementary materials: Please give caption for all figures and tables. Give the PCR products sizes.

The captions of supplementary material were included in the section called “Supplementary Material” at the end of manuscript before references section and also in the figure slide.

Supplementary Fig 3 is not mentioned in the text.

We included the Suppl. Fig. 3 in the section 3.3 of Results.